**Subject Category:**
Biology (whole organism)

behaviour

social communication, *Parus major*, mobbing, food-associated vocalizations

**Author for correspondence:**
Nadine Kalb
e-mail: nadine.kalb@uni-tuebingen.de

# Great tits encode contextual information in their food and mobbing calls

## Nadine Kalb, Fabian Anger and Christoph Randler

Department of Biology, Eberhard Karls Universität Tübingen, Auf der Morgenstelle 24, 72076 Tübingen, Germany

NK, 0000-0001-8318-5255

The calling behaviour of Paridae species (i.e. titmice, tits and chickadees) in a predator-related context is well-studied. Parid species are known to alter call types, note composition or call duration according to predation risk. However, how these species encode information about a non-threatening context, such as food sources, has been subject to only few studies. Studies in Carolina chickadees (*Poecile carolinensis*) have shown that this species alters the ratio of C and D notes to encode information about the presence of food and/or the flight behaviour of the signaller. This suggests that parids also use graded signals to encode information about non-predatory contexts. No study to date has directly compared the calls of a feeding context with those of a predation (i.e. mobbing) context. Hence, the aim of this study was to compare the calling behaviour of these two situations in great tits (*Parus major*). Calls uttered at a feeder were recorded, analysed and compared with calls uttered in front of taxidermy mounts of sparrowhawks (*Accipiter nisus*). In the food context, great tits reduced the number of D notes and increased the number of B, C and E notes compared with the mobbing context. Furthermore, tits produced calls with longer D notes and shorter intervals between D notes than in the mobbing context. This indicates that great tits use two mechanisms of graded signals (i.e. note type and acoustic structure of D calls) to inform conspecifics about the nature of a situation.

## 1. Introduction

There is a growing body of research about if and how animals use vocalizations to encode information about their environment (reviewed in [1]). Calling behaviour is especially well studied in a predation context across various taxa. However, calling can hold costs, as it can, for example, increase predation risks by revealing the location of the caller or nest to the predator [2–6]. Nonetheless, producing anti-predator vocalizations can also be beneficial, as it might drive the predator away, warn conspecifics or signal the

predator that it has been detected and consequently decrease its hunting success (pursuit deterrent signal) [7–9]. Consequently, even if calling might hold costs, transmitting information about the environment to con- and heterospecifics can increase the (long-term) fitness of the caller. Information can be encoded referentially or graded. Referential calls are discrete calls and transmit information about an event (e.g. predation) or object (e.g. food) and thereby enable the receiver to show an appropriate behavioural response without any additional cues [10]. Many studies have shown that animals use referential calls in various situations. In mammals, calls are used in a predation context, after food discovery and during social interaction [11]. Vervet monkeys (*Cercopithecus aethiops*), for example, produce different alarm calls for leopards (*Panthera pardus*), eagles and snakes whereby each alarm call elicits a predator-specific response [12]. Red squirrels (*Tamiasciurus hudsonicus*; [13] and suricates (*Suricata suricatta*; [14,15]) produce distinct calls for terrestrial and aerial predators. Marmosets (*Callithrix geoffroyi*) produce distinct food calls and increase feeding and foraging rates after hearing such calls [16]. In birds, Japanese great tits (*Parus minor*) produce distinct calls in response to rat snakes (*Elaphe climacophora*), jungle crows (*Corvus macrorhynchos*) and martens (*Martes melampus*; [17,18]). Further, fledglings [17] and adults [19,20] show different anti-predator behaviours after hearing such calls. Some species, such as the domestic chicken also use referential calls in a food-associated context [21].

In contrast to referential signals, graded signals do not use different call types, but changes in calling rate or call structure to encode information. Many birds and mammals, for example, change calling rate in a food- or predator-related context [22–27]. Great tits (*Parus major*) increase calling rate when predation risk increases [27,28], whereas crested tits (*Parus cristatus*) adjust the calling rate of their calls to changes in habitat safety [29]. Besides calling rate, birds have been shown to alter the duration of notes or interval between notes [30,31], as well as call propensity or the proportion of different call types or a combination of those parameters to encode information about predator threat [27,32]. Tufted titmice (*Baeolophus bicolor*) and black-capped chickadees (*Poecile atricapillus*) produce calls with more D notes in front of more threatening predators [30,33], and Carolina chickadees (*Poecile carolinensis*) alter the ratio of 'chick' and 'dee' notes according to predation risk [34]. Great tits alter the number of D notes, as well as the interval between them to discriminate between two common predators [31]. During playback experiments, great tits responded differently concerning different call rates and D call structure [35,36], suggesting that graded signals convey information about the urgency of a threat. In chaffinches (*Fringilla coelebs*), con- and heterospecifics responded with a nearer approach when confronted with the same call when play backed in a higher duty cycle [26].

While the acoustic structure of predator-related calls is well studied, studies on the acoustic structure of food-associated calls are rather scarce. Chimpanzees (*Pan troglodytes*) have been shown to alter the acoustic structure of food calls according to food preference with highly preferred foods eliciting longer calls with higher peak frequencies [37]. In birds, most studies focused on the proportion of different call or note types in a feeding context. Willow tits (*Poecile montanus*), for example, use calls of a single note type in a food context and combine two distinct call types in a non-food context [38]. Moreover, a playback study revealed that calls from a feeding context attract con- and heterospecifics and hence most likely serve a recruitment function [39]. The *chick-a-dee* call of the genus *Poecile* consists of up to four note types (A, B, C and D) and birds seem to encode information by altering the note composition and repetition of note types [40]. Black-capped chickadees, for example, produce *chick-a-dee* calls when finding a food source to attract flock members [41]. Carolina chickadees produce calls containing a high proportion of C notes after finding food and are more likely to visit a feeding station after hearing playbacks with a large number of C notes [42]. Mahurin & Freeberg [43], in contrast, showed that Carolina chickadees in another population that initially found a food source produced calls with a higher number of D notes before a second individual arrived and birds arrived faster at a feeding site after hearing calls with a greater number of D notes.

Since both anti-predator behaviour and finding food are essential for an individual's fitness, but significantly differ in their nature (i.e. dangerous versus non-dangerous), it would be beneficial for birds to encode contextual information in their calls. However, to our best knowledge, no study to date has directly compared calls from a feeding context with calls from a predation context. Hence, the aim of this study was to examine if the calls of great tits in a mobbing context differ from those in a feeding context. Based on previous work discussed in the introduction, we hypothesized that the proportion of call types uttered in a mobbing context differs from calls in a feeding context. A recent study found great tits to alter the duration of D calls as well as the number of elements and the interval between them according to different predatory contexts [31]. Therefore, we expected tits to also alter the acoustic structure of D notes between the mobbing and the feeding context. Calls of wild great tits were recorded in two experimental situations (presentation of food or a predator mount) to test our hypotheses.

# 2. Material and methods

## 2.1. Study species and sites

All experiments were conducted on wild great tits (*P. major*) in the vicinity of Tübingen, (48°31′ N, 9°3′ E) and Rottenburg am Neckar (48°28′ N, 8°56′ E), Baden-Württemberg in southwest Germany. The study was performed during the non-breeding season of great tits in January–March (2017, 2018, 2019) and August–December (2017, 2018). The same location was not visited more than once and stimuli presentations were always separated by a minimum distance of 214 m (mean ± s.e., 439.76 m ± 59.32 m). This distance can be considered to be sufficient to ensure independent measures as a minimum distance of 200–250 m is also used in other studies to perform independent observations of free-ranging parids [42,44]. Additionally, in the present study area, the density of great tits is very high, for example, at the Spitzberg between Tübingen and Rottenburg, there are about 370–390 pairs on 623 ha [45]. Hence, even though great tits in this study were not individually ringed, the likelihood of testing an individual twice was low. Since birds could not be individually identified, only one stimulus per site was presented [food, sparrowhawk or green woodpecker (*Picus viridis*) mount] and each location was treated as an independent sample unit. Observations took place between 07.30 and 16.00 CET to allow birds to recover from or prepare for the night to reduce the stress on birds. During all trials, the observer kept a minimum distance of 8 m to the stimulus. All calls were recorded with a digital recorder (Marantz professional PMD661MKIII, inMusic GmbH, Ratingen, Germany). In 2017 and January–March 2018 a shotgun microphone (Elektret K6/ME66, Sennheiser electronic GmbH, Wedemark, Germany) was used to record food calls. During all other sampling periods, the recorder was connected to a boundary microphone (Marantz professional, inMusic GmbH, Ratingen, Germany).

## 2.2. Predator context

Two different mounts of sparrowhawks (*Accipiter nisus*) were used to elicit mobbing calls to reduce pseudo-replication and only one sparrowhawk mount was presented per site. Ideally, one would present a different mount at each location to ensure the statistical independence of all recordings. However, Beránková *et al*. [46] showed that a single dummy is sufficient for great tits to recognize a sparrowhawk as long as it displays the local key features of the species. The mounts used in the present study did not greatly differ from each other in size, coloration or shape and are considered to display all important predator features. Additionally, a previous study investigating mobbing vocalizations with these specimens did not reveal a difference in great tit behaviour between the two predator mounts [31]. The mounts were placed on tree trunks, fences or rocks approximately 150–200 cm above the ground. Sparrowhawks are common predators on small songbirds [47] and breed widespread in this area, with a total of 10–20 pairs in the surrounding (personal observation). Furthermore, great tits are known to have lower body masses and reduce feeding periods when sparrowhawks are abundant in the area [48–50]. Hence, great tits seem to perceive sparrowhawks as high-threat predators, which makes them well suited for our study to provoke mobbing calls.

The location, mount identity and time were documented at the start of each recording. The microphone was placed next to the mount and recordings started immediately after setting up the equipment. Recordings were terminated 10 min after great tits arrived at the study location, in cases where no great tit participated in mobbing, observations were terminated after 30 min. The observer noted the number of conspecifics in a radius of 5 m around the taxidermy mount. In total, predator presentations at 29 different locations were conducted.

## 2.3. Food calls

Two weeks prior to sound recordings, a hanging feeder was placed in a tree 2–3 m above the ground. Ten different PVC feeders, all from the same type (dobar Art. 7948357, Germany) were used throughout the study. Each feeder was stocked with black oil sunflower seeds every 4–5 days to get the birds accustomed to the feeders as an irregular food source.

Upon recording, the feeder was fully stocked, and the recording equipment was set up. Recordings started immediately after setting up the equipment and were terminated 30 min after the first great tit visited the feeder. In cases where no great tit visited the feeding station, recordings were terminated after 60 min. The maximum observation period in feeding trials was longer than during predator

presentations as it usually takes longer for birds to visit a feeder and utter calls than to start mobbing (personal observations). Hence, the longer observation period during feeding trials was necessary to reliably determine the behaviour of the first great tit at the feeder. The observer noted whether a great tit visited the feeder and whether it called or not. Further, if a great tit visited the feeder, the number of conspecifics in a radius of 5 m around the feeder was noted. In total, feeders were installed at 35 independent locations, but only in 24 of those locations, great tits visited the feeder and took at least one sunflower seed. At 18 locations, great tits called while visiting the feeder.

## 2.4. Control experiments

To test if great tits respond to a specific stimulus (i.e. food or sparrowhawk) or simply utter calls in response to the presence of any stimulus, birds were additionally confronted with two mounts of European green woodpeckers (*P. viridis*) as a negative control at 16 independent locations. For woodpeckers, only one mount was presented per site. Mounts were placed on tree trunks, fences or rocks approximately 150–200 cm above the ground. The population density of the green woodpecker is even higher than that of the sparrowhawk with an estimate of 50–100 pairs (see [45] (Spitzberg area), Randler 2018, unpublished data (Weggental/Rottenburg)). Green woodpeckers are well known and overlap in their habitats with great tits but pose no danger, nor are they competitors for food because green woodpeckers mainly feed on ants (usually *Lasius* sp.) [51].

## 2.5. Permission to carry out fieldwork

The study was conducted in accordance with the higher nature conservation authority in Tübingen and therefore not required to complete an additional ethical assessment prior to conducting the research.

## 2.6. Call analysis

Files were analysed using Avisoft SASLabPro with a sample rate of 44.1 kHz. First, sonograms using the Hamming window function, FFT length 512, frame size 100% and 75% overlap were created to determine all calls produced in the respective contexts (feeding and mobbing). All calls during the first 3 min after great tits started to utter calls were analysed. Because, to our best knowledge, there is no specific description of note types used during mobbing and feeding for great tits, the description of note types in a closely related species, the Japanese great tit (*P. minor*), was used as a guideline to classify notes [52]. Furthermore, the number of notes per call was counted.

An additional sonogram (Hamming window function, FFT length 1024, frame size 25% and 98.43% overlap) was created to perform more fine-scale measurements of the first five D calls, i.e. the duration (s) of calls and notes within a call as well as the interval (s) between notes. For this analysis, only those calls recorded with the omnidirectional microphone were evaluated to exclude any effects of the recording equipment on the measured call features.

## 2.7. Statistical analyses

Principal component analyses (PCA) were performed to analyse the acoustic features of calls composed of D notes. Two of the four principal components had an eigenvalue greater than one (PC1: 1.998, PC2: 1.61) and hence complied with the Kaiser criterion [53]. PC1 and PC2 explained 88.8% of the total variance. Factor scores derived from the PCA were used as acoustic features of the D calls for further analysis. ANOVAs including PC1 and PC2 as response variable and context, as well as number of great tits and heterospecific individuals as fixed factors, were conducted. For all tests, the significance level was set to $\alpha = 0.05$. For comparison between contexts, the mean and the standard error are given.

# 3. Results

In 20 out of 29 locations, great tits participated in mobbing. However, in four locations, great tits did not produce mobbing calls. During all control experiments ($n = 16$), great tits were present in a radius of 10 m around the woodpecker mounts. However, tits did not approach the control within a radius of 5 m nor did they utter calls. In the mobbing and food context, nine different note types were recorded (figure 1). As great tits did not call during control trials but responded to food and the sparrowhawk mount, one

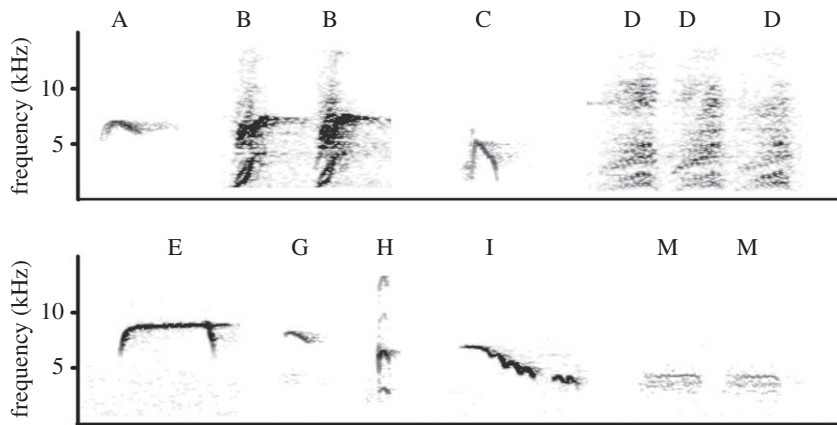

**Figure 1.** Spectrogram illustration of the nine different note types that great tits produced during mobbing or feeding. The top frame illustrates note types A–D, whereas the bottom frame depicts notes E–M (left to right). Hamming window function FFT length 512, frame size 100%, 75% overlap. Noise below 1 kHz was removed.

**Table 1.** Percentages of note types uttered by great tits in response to food and sparrowhawk.

|   | food | mobbing |
|---|------|---------|
| A | 3.83 | 0.44 |
| B | 15.74 | 9.05 |
| C | 13.19 | 0.38 |
| D | 41.7 | 85.42 |
| E | 14.47 | 4.02 |
| G | 0.85 | 0.00 |
| H | 2.13 | 0.00 |
| I | 5.11 | 0.69 |
| M | 2.98 | 0.00 |

can conclude that great tits reacted to the specific stimulus (i.e. food or predator) rather than just the presence of any stimulus.

The proportion of note types given by great tits differed significantly between contexts (likelihood ratio $\chi^2 = 287.28$, d.f. = 8, $p < 0.0001$). Among the six note types given in both context, D notes were the most common ones (table 1). In the mobbing context, great tits produced mainly D notes and small percentages of A, B, C, E and I notes. G, H and M notes were solely produced in the food context. Further, great tits produced a smaller percentage of D notes and an increased percentage of A, B, C and E notes compared with the mobbing context (table 1). Call rate (calls/minute/individual) was significantly affected by context ($F = 42.544$, d.f. = 1,1, $p < 0.0001$) and the number of conspecifics ($F = 10.027$, d.f. = 1,1, $p = 0.004$), but not the number of heterospecific individuals ($F = 1.232$, d.f. = 1,1, $p = 0.2766$). Great tits produced more calls per minute in the mobbing context (16.62 ± 2.84) than in the food context (3.25 ± 0.73).

In respect of acoustic features, we found a significant difference between contexts. PC1 explained 49.9% of the total variance and correlated strongly with the number of D notes in a call and call duration. PC2 explained 40.3% of the variance and correlated strongly with both note duration and interval between notes (table 2).

According to the loading coefficients, high scores of the principal components translate into a stronger response (i.e. high number of notes, longer calls, longer notes and intervals between notes). The PC2 scores were significantly affected by context ($F = 30.52$, d.f. = 1,1, $p < 0.0001$) whereas PC1 scores were not ($F = 0.019$, d.f. = 1,1, $p = 0.9653$). None of the PC scores were affected by the number of great tits or heterospecifics (all $p > 0.2$). Great tits produced calls with longer D notes and shorter

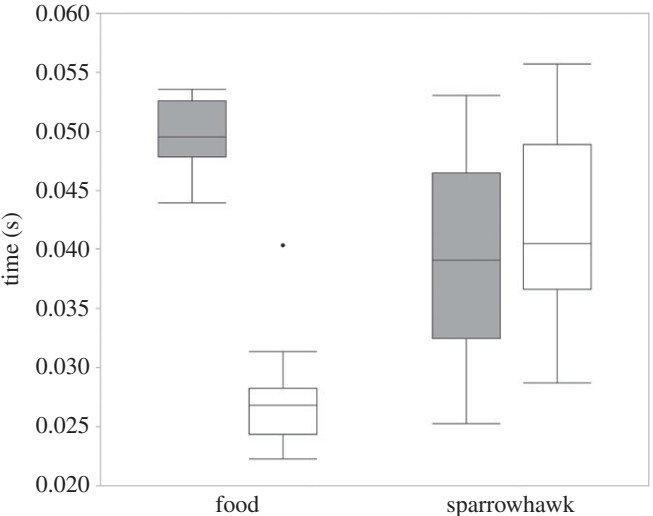

**Figure 2.** Mean duration of D notes (grey boxplots) and interval between notes (white boxplots) in seconds. D calls in response to food have longer notes and shorter intervals between notes than D calls in response to a sparrowhawk mount.

**Table 2.** Eigenvalues, explained variances and loading of the mean acoustic measures of D notes on the two PC factors. Loadings in italic are larger than 0.50 or smaller than −0.50 (italic parameters have a strong relationship with the respective PC component).

|  | PC1 | PC2 |
| --- | --- | --- |
| eigenvalue | 1.997 | 1.61 |
| % variance | 49.927 | 40.259 |
| number of D notes | *0.964* | 0.17 |
| note duration | 0.339 | *−0.842* |
| call duration | *0.972* | 0.213 |
| interval between D notes | −0.094 | *0.91* |

**Table 3.** Mean values ± s.d. of four acoustic parameters of D calls in response to food and a sparrowhawk mount.

|  | food | mobbing |
| --- | --- | --- |
| number of D notes | 6.95 ± 0.434 | 7.13 ± 0.467 |
| note duration | 0.05 ± 0.002 | 0.039 ± 0.002 |
| call duration | 0.507 ± 0.033 | 0.541 ± 0.042 |
| interval between D notes | 0.028 ± 0.002 | 0.042 ± 0.008 |

intervals between notes in the food context compared with the mobbing context (figure 2 and table 3). Mean call duration and mean number of notes did not differ between contexts (table 3).

## 4. Discussion

Great tits are monogamous birds living in pairs during the breeding season, but form flocks consisting of adults and non-related juveniles of both sexes during the non-breeding season [54,55]. Being in a flock might, on one hand, reduce access to food due to competition, but can, on the other hand, hold the benefit of reduced predation risk due to dilution effects. Moreover, flocks might be able to detect food or approaching predators faster than single individuals. Nonetheless, those possible advantages

require sufficient communication between flock members about the environment. By encoding information in vocal signals, flock members are able to communicate with each other and coordinate their behaviour such as mobbing [56] or foraging [57].

In our study, we found a significant difference between calls of a food-associated and mobbing context. Great tits produced mostly D notes and a small percentage of other notes in the mobbing context. In the feeding context, by contrast, great tits decreased the proportion of D notes and increased the proportion of other notes. This suggests that the ratio of D notes to other notes might be used to differentiate between mobbing and feeding situations. Great tits produced significantly more D calls in the mobbing context than in the feeding context, which is in line with previous findings in parids where calling rate and the proportion of notes changes with increasing predator threat [27,30]. Additionally, great tits in the present study produced a small proportion of G, H and M notes solely in the feeding context, indicating that note composition might convey information about food availability.

Even though in our study G, H and M notes solely occurred in the food-associated context, we would not claim them to be signals exclusively used in a food context by great tits. Japanese great tits, a closely related species, are known to produce those notes also in a predation context [52]. Hence, one can assume that great tits also might use those note types in a mobbing context that differs from the situation simulated in the present study, for example, when confronted with a different predator than the sparrowhawk.

Similar to our results, Freeberg & Lucas [42] found Carolina chickadees to produce calls containing a higher proportion of C notes when detecting a food source. Moreover, birds approached a feeder more frequently after hearing C-rich playbacks than after hearing calls containing no C notes (long D calls), indicating that C notes might convey information about the presence of food. However, a second study in a different population of Carolina chickadees found individuals to produce more D notes when first finding a food source, suggesting that D notes in this species have a general recruitment function [43]. Results of a second study in the same population suggested that C notes are associated with flight, as birds that were flying produced more C notes than when e.g. sitting on a perch [58]. Freeberg & Mahurin suggested that these varying results might be explained by D notes being recruitment calls and C notes stimulating flight behaviour in receivers, which are in turn more likely to find food as they move through the environment. This might also be an explanation for our results, as in a feeding context, birds moved from and to the feeder, resulting in a higher proportion of C notes. In the mobbing context, by contrast, birds were more restricted to the area around the taxidermy mount and tried to recruit con- and heterospecifics, resulting in an increased D call production. Hence, we propose that great tits might be able to gain contextual information about the nature of a situation by the ratio of D to other notes. While few to no C and E notes and a high number of D notes might encode a predation context, a more balanced ratio of E and C notes and fewer D notes might encode the presence of food. Playback studies are crucial to determine if great tits alter their behaviour in response to conspecific calls of a food and mobbing context. Here, future studies might alter the ratio of notes and measure the latency time until great tits arrive at a feeding station, similar to the studies by Freeberg & Mahurin [43,58].

In addition to variation in note types, great tits in our study might have gained additional information by subtle variations in D calls as they significantly differed between contexts. Food-associated calls had shorter intervals between notes than mobbing calls. That mobbing calls have longer intervals compared with food-associated calls is similar to findings in great tits, showing that tits have longer intervals in response to sparrowhawks than when seeing a less dangerous tawny owl mount [31]. Templeton *et al.* [30] also found black-capped chickadees to alter the interval between the first and second D note, whereby the interval was shorter in response to more dangerous predators. This indicates that a variation in the interval between notes might be used by various passerines to encode information about different contexts, and more studies are needed to investigate if and how this mechanism is used during communication. Great tits seem to be able to recognize such subtle variation in mobbing calls, as they behave differently when hearing conspecific mobbing calls provoked by different predators [36].

Moreover, food-associated calls were found to have longer D notes compared with those produced in response to the sparrowhawk mount. This is in contrast to findings in black-capped chickadees, which decrease the duration of the first D note when confronted with smaller, more dangerous predators [30]. Nonetheless, this is most likely explained by the fact that in Templeton's study, also the number of notes and call duration increased when the note duration and interval between notes decreased. In our study, however, call duration and number of notes did not differ. Further, a previous study in great tits comparing mobbing calls in response to two predators did not find a variation in note duration [31]. This suggests that great tits might use D note duration to discriminate between a predation and a non-predatory context (e.g. feeding) and note number, interval between notes as well

as calling rate is used to encode information about threat level [27,30,31]. However, a playback study which experimentally manipulates those parameters is necessary to test this assumption.

Lastly, the present study did not investigate if great tits might also encode contextual information by a variation in frequency, bandwidth or call entropy similar to other species [30,59]. A previous study revealed that great tits do not use the frequency or bandwidth of D notes to encode information about different predators [31]. Nonetheless, these parameters might be used to differentiate between a feeding and a predation context similar to D note duration. Hence, future studies should also investigate these possible ways of encoding information in passerine birds.

Our results showed that great tits alter the fine-scale acoustic structure of D calls according to a different context. Moreover, the proportion of notes differed between a food-associated and a mobbing context, which indicates that tits, similar to other species, might use the ratio of notes to discriminate between different contexts. Future studies are needed to determine if also other parids use these mechanisms of encoding information and further, if con- and heterospecifics alter their behaviour in response to such calls.

Ethics. The study was not required to complete an ethical assessment prior to conducting the observations. It was performed in accordance with relevant guidelines and regulations for nature conservancy in Germany (§44 Abs. 1 Nr. 2 BNatSchG) and adhered to the Guidelines for the Use of Animals in Research of the Animal Behavior Society/Association for the Study of Animal Behaviour.
Data accessibility. The datasets generated and analysed during the study are available on the Dryad Digital Repository: https://dx.doi.org/10.5061/dryad.h7k7551 [60].
Authors' contributions. N.K. and C.R. designed the experiments, N.K. and F.A. collected the field data. N.K. analysed the bioacoustics results and did the statistical analysis. All authors contributed to the writing of the paper and have approved its final stage.
Competing interests. All authors declare no conflict of interest.
Funding. This work was supported by the Gips-Schüle-Stiftung.

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
