## [Reviewer comments · Royal Society Open Science]

Review History

RSOS-191210.R0 (Original submission)

Review form: Reviewer 1

Is the manuscript scientifically sound in its present form?

Yes

Are the interpretations and conclusions justified by the results?

Yes

Is the language acceptable?

Yes

Do you have any ethical concerns with this paper?

No

Have you any concerns about statistical analyses in this paper?

No

Recommendation?

Accept with minor revision (please list in comments)

Comments to the Author(s)

Kalb et al. describe a series of experiments where great tits were presented with either food or a mount of a sparrowhawk. The question was whether the syntax of the calls follows patterns already established in the literature. The novelty of the current study is presumably comparing the response to both stimuli (albeit used separately) in the same study.

The study shows that what has been written about this call system to date is correct. The direct comparison isn't exactly helpful in itself, but with this study we can now add great tits to the list of species that encode both mobbing and foraging contexts in the chick-a-dee call.

The paper is reasonably well written, and the stats and data are believable.

One minor note is that mobbing conditions are usually generated using a less dangerous predator than the sparrowhawk – certain owls for example. I guess the tits produce mobbing calls instead of alarm calls because the predator is sitting. Surely they would not mob a flying sparrowhawk, would they? Were there any incidents where a single or few alarm calls were given? Along these lines, were there any differences in the response to the different mounts? Finally, the entropy level of the D notes should be measured along with duration and inter-note intervals with the obvious predation that birds should use less entropic D notes in a foraging context.

Review form: Reviewer 2

Is the manuscript scientifically sound in its present form?

Yes

Are the interpretations and conclusions justified by the results?

Yes

Is the language acceptable?

Yes

Do you have any ethical concerns with this paper?

No

Have you any concerns about statistical analyses in this paper?

No

Recommendation?

Accept with minor revision (please list in comments)

Comments to the Author(s)

Review of: RSOS-191210 Great tits encode contextual information in their food and mobbing calls

This is an interesting study that explores the vocal behaviour of Great Tits in two contexts, feeding and predation/mobbing. Here, the authors sought to compare chick-a-dee mobbing calls produced in these two contexts, something not yet addressed in the literature. The results are

clear and important, demonstrating that D note length and internote interval vary based on context. However, there are a few issues the authors should consider when preparing a revision:

Line 8: Change to "(i.e., titmice, tits, and chickadees)".

Line 16: Add a comma following each mention of "i.e.", here and throughout the document.

Line 30: Add semicolons between citations, here and throughout the document.

Line 37: Add a comma following each mention of "e.g.", here and throughout the document.

Line 44, 49: Remove the middle brackets and replace with a semicolon.

Line 85: Italicize "chick-a-dee" throughout to be consistent with the literature.

Line 101: Remove the period before the in-text citation.

Lines 109-110: The authors should clarify that these are non-breeding seasons.

Line 110-111: Reword to "The same location was not visited more than once,..."

Line 111: Is this enough distance compared to the territory of Great tits to ensure that you weren't exposing conditions to the same individual more than once?

Line 112: Change "e.g." to "for example".

Lines 122-124: There are too many brackets back-to-back. Please refine.

General comment: 'We' should be removed throughout the document and reworded to passive voice. For example, "We used two different mounts of sparrowhawks..." (line 126) should be changed to "Two distinct mounts of sparrowhawks were used...". These errors primarily occur in the Results section.

Line 126-127: Was there no way to acquire and use more mounts? How representative are the two mounts used? I am sympathetic to this problem but would feel better with some assurance that there was nothing particularly odd or distinctive about these mounts.

Line 129: The size of the text for the citation appears to be in a smaller font.

Line 144: The closing bracket should not be italicized.

Line 146-147: Why did the authors use one criteria for determining trial length in mount exposure (terminated the trial 10 minutes after the first Great Tit arrived for the mount exposure) and food trials (terminated 30 minutes after the setting up equipment)? Some rationale and explanation is needed here.

Lines 154-164: The negative control of the woodpecker is a good idea. Did the authors consider using an inanimate object, such as a block, as an additional control? Previous experiments have also used foam controls.

Line 183: Statistics needs to be plural here, or changed to Statistical Analyses.

Lines 225-228. This sentence runs on and requires some clarity. Also, keep the mention of “food” then “predator” in this parallel order to be consistent with the first part of the sentence and the rest of the document.

Line 247: Change "e.g." to "for example".

Line 257, 266: What is the purpose of the use of "e.g." here?

Line 285: Change to "...food-associated calls to have longer D notes compared to those produced in response to the sparrowhawk mount".

References: Some journal titles are italicized, whereas others are not.

Figure 1: This caption should clarify that notes A-D are pictured in the top frame, and notes E-M are pictures in the bottom frame. As for the figure itself, it's missing the x-axis of Time (ms?).

Decision letter (RSOS-191210.R0)

03-Sep-2019

Dear Miss Kalb

On behalf of the Editors, I am pleased to inform you that your Manuscript RSOS-191210 entitled "Great tits encode contextual information in their food and mobbing calls" has been accepted for publication in Royal Society Open Science subject to minor revision in accordance with the referee suggestions. Please find the referees' comments at the end of this email.

The reviewers and handling editors have recommended publication, but also suggest some minor revisions to your manuscript. Therefore, I invite you to respond to the comments and revise your manuscript.

- Ethics statement

- Data accessibility

<http://datadryad.org/submit?journalID=RSOS&manu=RSOS-191210>

- **Competing interests**

- **Authors' contributions**

- **Acknowledgements**

- **Funding statement**

Because the schedule for publication is very tight, it is a condition of publication that you submit the revised version of your manuscript before 12-Sep-2019. Please note that the revision deadline will expire at 00.00am on this date. If you do not think you will be able to meet this date please let me know immediately.

on behalf of Prof Kevin Padian (Subject Editor)
openscience@royalsociety.org

Associate Editor Comments to Author:

Two reviewers have provided an assessment of your paper, and each other a number of recommendations to improve the manuscript; however, it is the view of the Editors that these appear to be minor tweaks. Please ensure you provide a point-by-point response to their comments in your revision and also a tracked-changes version of the manuscript to help us identify the changes you have made. Good luck and thanks for your submission to RSOS!

Reviewer comments to Author:

Reviewer: 1

Comments to the Author(s)

Kalb et al. describe a series of experiments where great tits were presented with either food or a mount of a sparrowhawk. The question was whether the syntax of the calls follows patterns already established in the literature. The novelty of the current study is presumably comparing the response to both stimuli (albeit used separately) in the same study.

The study shows that what has been written about this call system to date is correct. The direct comparison isn't exactly helpful in itself, but with this study we can now add great tits to the list of species that encode both mobbing and foraging contexts in the chick-a-dee call.

The paper is reasonably well written, and the stats and data are believable.

One minor note is that mobbing conditions are usually generated using a less dangerous predator than the sparrowhawk – certain owls for example. I guess the tits produce mobbing calls instead of alarm calls because the predator is sitting. Surely they would not mob a flying sparrowhawk, would they? Were there any incidents where a single or few alarm calls were given? Along these lines, were there any differences in the response to the different mounts? Finally, the entropy level of the D notes should be measured along with duration and inter-note intervals with the obvious predation that birds should use less entropic D notes in a foraging context.

Reviewer: 2

Comments to the Author(s)

Review of: RSOS-191210 Great tits encode contextual information in their food and mobbing calls

This is an interesting study that explores the vocal behaviour of Great Tits in two contexts, feeding and predation/mobbing. Here, the authors sought to compare chick-a-dee mobbing calls produced in these two contexts, something not yet addressed in the literature. The results are clear and important, demonstrating that D note length and internote interval vary based on context. However, there are a few issues the authors should consider when preparing a revision:

Line 8: Change to "(i.e., titmice, tits, and chickadees)".

Line 16: Add a comma following each mention of "i.e.", here and throughout the document.

Line 30: Add semicolons between citations, here and throughout the document.

Line 37: Add a comma following each mention of "e.g.", here and throughout the document.

Line 44, 49: Remove the middle brackets and replace with a semicolon.

Line 85: Italicize "chick-a-dee" throughout to be consistent with the literature.

Line 101: Remove the period before the in-text citation.

Lines 109-110: The authors should clarify that these are non-breeding seasons.

Line 110-111: Reword to "The same location was not visited more than once..."

Line 111: Is this enough distance compared to the territory of Great tits to ensure that you weren't exposing conditions to the same individual more than once?

Line 112: Change "e.g." to "for example".

Lines 122-124: There are too many brackets back-to-back. Please refine.

General comment: 'We' should be removed throughout the document and reworded to passive voice. For example, "We used two different mounts of sparrowhawks..." (line 126) should be changed to "Two distinct mounts of sparrowhawks were used...". These errors primarily occur in the Results section.

Line 126-127: Was there no way to acquire and use more mounts? How representative are the two mounts used? I am sympathetic to this problem but would feel better with some assurance that there was nothing particularly odd or distinctive about these mounts.

Line 129: The size of the text for the citation appears to be in a smaller font.

Line 144: The closing bracket should not be italicized.

Line 146-147: Why did the authors use one criteria for determining trial length in mount exposure (terminated the trial 10 minutes after the first Great Tit arrived for the mount exposure) and food trials (terminated 30 minutes after the setting up equipment)? Some rationale and explanation is needed here.

Lines 154-164: The negative control of the woodpecker is a good idea. Did the authors consider using an inanimate object, such as a block, as an additional control? Previous experiments have also used foam controls.

Line 183: Statistics needs to be plural here, or changed to Statistical Analyses.

Lines 225-228. This sentence runs on and requires some clarity. Also, keep the mention of "food" then "predator" in this parallel order to be consistent with the first part of the sentence and the rest of the document.

Line 247: Change "e.g." to "for example".

Line 257, 266: What is the purpose of the use of "e.g." here?

Line 285: Change to "...food-associated calls to have longer D notes compared to those produced in response to the sparrowhawk mount".

References: Some journal titles are italicized, whereas others are not.

Figure 1: This caption should clarify that notes A-D are pictured in the top frame, and notes E-M are pictures in the bottom frame. As for the figure itself, it's missing the x-axis of Time (ms?).

Author's Response to Decision Letter for (RSOS-191210.R0)

See Appendix A.

Decision letter (RSOS-191210.R1)

01-Oct-2019

Dear Miss Kalb,

I am pleased to inform you that your manuscript entitled "Great tits encode contextual information in their food and mobbing calls" is now accepted for publication in Royal Society Open Science.

on behalf of Prof Kevin Padian (Subject Editor)
openscience@royalsociety.org

Appendix A

Associate Editor Comments to Author:

Two reviewers have provided an assessment of your paper, and each other a number of recommendations to improve the manuscript; however, it is the view of the Editors that these appear to be minor tweaks. Please ensure you provide a point-by-point response to their comments in your revision and also a tracked-changes version of the manuscript to help us identify the changes you have made. Good luck and thanks for your submission to RSOS!

Response: Thank you for considering our study for publication in your journal. We incorporated the suggestions of both reviewers into our manuscript and responded to each comment below.

Reviewer comments to Author:

Reviewer: 1

Comments to the Author(s)

Kalb et al. describe a series of experiments where great tits were presented with either food or a mount of a sparrowhawk. The question was whether the syntax of the calls follows patterns already established in the literature. The novelty of the current study is presumably comparing the response to both stimuli (albeit used separately) in the same study.

The study shows that what has been written about this call system to date is correct. The direct comparison isn't exactly helpful in itself, but with this study we can now add great tits to the list of species that encode both mobbing and foraging contexts in the chick-a-dee call.

The paper is reasonably well written, and the stats and data are believable.

Response: Thank you for reviewing our work. Your review helped us to improve our manuscript. We addressed all of your comments in more detail below.

One minor note is that mobbing conditions are usually generated using a less dangerous predator than the sparrowhawk – certain owls for example. I guess the tits produce mobbing calls instead of alarm calls because the predator is sitting. Surely they would not mob a flying sparrowhawk, would they? Were there any incidents where a single or few alarm calls were given?

Response: Thank you for these interesting questions. In our study we did not record any alarm calls. Great tits usually display mobbing behavior towards sparrowhawks both while they are perching and moving (personal observations), whereby the intensity of mobbing surely depends on the behavior of the predator. Alarm calls on the other hand are often associated with the presence of offspring (e.g., Suzuki 2011). We performed our study outside the breeding season, which might explain the lack of alarm calls. To investigate vocal communication in birds more recent studies have used various predator types (ranging from less threatening owls to highly threatening predators such as the sparrowhawk) (e.g., Templeton et al. 2005). For our study, we wanted to make sure that the difference in threat level between the two contexts (feeding and predation) is quite large, hence we decided to use a sparrowhawk to provoke mobbing in great tits.

Along these lines, were there any differences in the response to the different mounts?

Response: We used the same specimens in another study (Kalb et al. 2019, scientific reports) investigating mobbing vocalizations in great tits and did not find any difference in great tit behavior depending on which mount was presented. We included some more information about this aspect in Line 135-141.

Finally, the entropy level of the D notes should be measured along with duration and inter-note intervals with the obvious prediction that birds should use less entropic D notes in a foraging context.

Response: Thank you for this interesting idea! Unfortunately, we did not perform any frequency or entropy measurements for the recordings of this study. Nonetheless, it is an interesting hypothesis that should be investigated in future studies. We therefore included this idea in the discussion Line 319-325.

Reviewer: 2

Comments to the Author(s)

Review of: RSOS-191210 Great tits encode contextual information in their food and mobbing calls

This is an interesting study that explores the vocal behaviour of Great Tits in two contexts, feeding and predation/mobbing. Here, the authors sought to compare chick-a-dee mobbing calls produced in these two contexts, something not yet addressed in the literature. The results are clear and important, demonstrating that D note length and internote interval vary based on context. However, there are a few issues the authors should consider when preparing a revision:

Response: thank you for helpful comments and suggestions, which helped us to improve the quality of our manuscript.

Line 8: Change to "(i.e., titmice, tits, and chickadees)".

Response: changed

Line 16: Add a comma following each mention of "i.e.", here and throughout the document.

Response: changed

Line 30: Add semicolons between citations, here and throughout the document.

Response: We added semicolons between all citations throughout the manuscript.

Line 37: Add a comma following each mention of "e.g.", here and throughout the document.

Response: we added commas after each e.g. throughout the manuscript.

Line 44, 49: Remove the middle brackets and replace with a semicolon.

Response: we made the suggested changes to Line 44 and 49.

Line 85: Italicize "chick-a-dee" throughout to be consistent with the literature.

Response: we incorporated the suggested changes in the manuscript.

Line 101: Remove the period before the in-text citation.

Response: removed

Lines 109-110: The authors should clarify that these are non-breeding seasons.

Response: we reworded this line to clarify that the experiment was performed outside the breeding season of great tits

Line 110-111: Reword to "The same location was not visited more than once,..."

Response: we reworded the lines as suggested.

Line 111: Is this enough distance compared to the territory of Great tits to ensure that you weren't exposing conditions to the same individual more than once?

Response: we explained in more detail in Line 115-118 why we consider this distance to be sufficient to ensure independent sampling.

Line 112: Change "e.g." to "for example".

Response: changed

Lines 122-124: There are too many brackets back-to-back. Please refine.

Response: we reworded this section to remove some of the brackets.

General comment: 'We' should be removed throughout the document and reworded to passive voice. For example, "We used two different mounts of sparrowhawks..." (line 126) should be changed to "Two distinct mounts of sparrowhawks were used...". These errors primarily occur in the Results section.

Response: we reworded to passive voice as suggested

Line 126-127: Was there no way to acquire and use more mounts? How representative are the two mounts used? I am sympathetic to this problem but would feel better with some assurance that there was nothing particularly odd or distinctive about these mounts.

Response: Ideally, one would of course use one mount per study site to ensure independent measures. However, it was not possible for us to acquire more mounts. Nonetheless, the mounts did not greatly differ in size, coloration or any other features. Hence, we considered them to be sufficient to provoke natural mobbing behavior in great tits. Moreover, we used the same specimens in another study (Kalb et al. 2019, scientific reports) and did not find any difference in great tit behavior depending on which mount was presented. We agree that it is an important issue that should be mentioned in the manuscript. Therefore, we included some more information about this aspect in Line 135-141.

Line 129: The size of the text for the citation appears to be in a smaller font.

Response: changed

Line 144: The closing bracket should not be italicized.

Response: changed

Line 146-147: Why did the authors use one criteria for determining trial length in mount exposure (terminated the trial 10 minutes after the first Great Tit arrived for the mount exposure) and food trials (terminated 30 minutes after the setting up equipment)? Some rationale and explanation is needed here.

Response: We terminated the predator presentations after 10 minutes to reduce the stress for participating birds. Since we considered the feeding context to be experimental situation, which does not pose stress to great tits, these trials were longer. Moreover, from personal experience, it takes longer for birds to visit a feeder (approx. 10-20 minutes) than to start mobbing (approx. 1-5 minutes). Hence, the longer observation period during feeding trials was necessary to reliably cover the behavior of the first great tit which arrives at the feeder. We explained why the observation periods differed between the contexts in Line164-184

Lines 154-164: The negative control of the woodpecker is a good idea. Did the authors consider using an inanimate object, such as a block, as an additional control? Previous experiments have also used foam controls.

Response: Thank you for this suggestion. Presenting an inanimate object as a control is certainly a good method to investigate the behavior of birds in a “neutral” condition. However, in a brief pilot study (5 trials) in which we presented a small plastic box to great tits we could not observe any bird approaching the object nor did they emit any calls (similar to the woodpecker treatment). Therefore, we did not further pursue this additional experimental condition.

Line 183: Statistics needs to be plural here, or changed to Statistical Analyses.

Response: We changed it to Statistical Analyses.

Lines 225-228. This sentence runs on and requires some clarity. Also, keep the mention of “food” then “predator” in this parallel order to be consistent with the first part of the sentence and the rest of the document.

Response: thank you for this helpful comment. We rephrased this section to make it clearer and easier to read.

Line 247: Change "e.g." to "for example".

Response: changed

Line 257, 266: What is the purpose of the use of "e.g." here?

Response: We removed the “e.g.” as it has no purpose in both sentences.

Line 285: Change to "...food-associated calls to have longer D notes compared to those produced in response to the sparrowhawk mount".

Response: We incorporated your suggestion and additionally changed the sentence to passive voice

References: Some journal titles are italicized, whereas others are not.

Response: thank you for spotting this! We corrected the respective titles to be consistent throughout the reference list.

Figure 1: This caption should clarify that notes A-D are pictured in the top frame, and notes E-M are pictures in the bottom frame. As for the figure itself, it's missing the x-axis of Time (ms?).

Response: We added the information about which notes are displayed in the top and bottom frame. Also thank you for commenting on the missing x-axis. By creating this graph we focused on illustrating the structure and frequency of each call type. Therefore, we decided to not include time on the x- axis as the displayed note types were not uttered in one sequence but extracted from different sound files for the purpose of this graph (but off course we used the same settings for each sonogram). By doing so, it would be difficult to include a continuous time axis in the figure, which reliably displays the natural intervals between single note types.